# AN EFFECTIVE VARIANCE CHANGE DETECTION METHOD UNDER CONSTANTLY CHANGING MEAN

## ABSTRACT

Effectively evaluating the viability of a procured organ in the transplant patient prior the procedure is of critical importance. Current viability assessment methods rely on evaluating the organ's morphology and/or laboratory biopsy results with limited effectiveness. A recently proposed, well-designed noninvasive method evaluated the viability status of organs by detecting the variance change point of their surface temperature through exploring the entire data profile. However, most part of the data in a temperature profile barely contains the change information, which yields a waste of computational resources of their method. This paper proposes an accelerating algorithm with a well-designed dual control windows scheme that can be extended to online change detection. The proposed method significantly improves the computational speed and retains the same change detection power as the method Gao19 through the removal of redundant data. Simulation and application results demonstrate the robust performance of the proposed method.

## 1 INTRODUCTION

Organ transplantation is an effective method to treat end-stage diseases of various organs. Due to the limitation of storage and transportation techniques, the viability of organs cannot be fully guaranteed. Therefore, the quality of procured organs must be evaluated before transplantation. Different organs can accept different cold ischemia time. For example, the upper limit of the cold ischemia time of the liver is about 12 hours, and that of the heart is only 8 hours. Therefore, in an organ transplantation operation, both doctors and patients are racing against time. The traditional evaluation often relies on subjective experience in the viability of the organ based on the physical shape and some functional data. In many cases, it is impossible to clearly distinguish the viable organs from nonviable organs. Laboratory biopsy usually cut off a small piece of the organ for an integrity test of the transplanted organs. However, the test results of samples cannot represent the viability state of the entire organ, which leads to a missing identification of the transplantable organ areas, and results in a waste of valuable organs. Therefore, in order to further improve the utilization rate of transplanted organs, a more accurate and convenient organ viability change detection method is desired.

It has been proved that there is a strong correlation between surface temperature of organs and their viability (Skowno & Karpelowsky, 2014; Vidal et al., 2014; Kochan et al., 2015), where the surface temperature of the viable organ fluctuates greatly. A research team of clinical scientists and engineers at Virginia Tech designed a set of noninvasive organ viability evaluation methods (Bhonsle et al., 2016; O'Brien et al., 2017; Gao et al., 2019). A porcine liver was used in the experiment. The organ surface was divided into a dense grid covering the entire liver surface. A noninvasive, high-precision thermal imaging system was used to measure the surface temperature of the liver. Temperatures were collected every ten minutes for 24 hours. Figure 1 is the temperature profile of a randomly selected spot on the organ surface. From the plots we see that the mean trend of the temperature changes slowly and smoothly in the perfusion process. The surface temperature fluctuates strongly in the first 12 hours, which indicates a high viability of the organ. After 12 hours' perfusion, a sudden viability drop appears, and the organ gradually loses its viability. The red vertical line is the potential viability change point. Then, the viability evaluation of a procured organ is transformed from a medical problem into a statistical problem of variance change point detection under a smoothly changing mean trend.

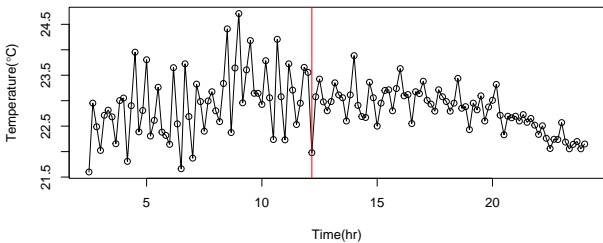

Figure 1: Temperature profile at a randomly selected spot of the liver. The red vertical line is the approximated location of the variance change point.

Recent research on the change point detection are mainly based on the parametric methods and the nonparametric methods. The parametric method assumes that the data distribution belongs to a certain distribution family, and infers sudden changes in one or several parameters of the data distribution (Inclan & Tiao, 1994; Chen & Gupta, 1997; Pan & Chen, 2006; Kang et al., 2018; Dette & Gösmann, 2020; Wang et al., 2021). However, these existing methods cannot be applied in our liver experiment, since not only the variance change suddenly, the mean function of the temperature profile also changes smoothly. The nonparametric change point detection method does not depend on an overall distribution (Hariz et al., 2007; Matteson & James, 2014; Zou et al., 2014; Yu & Chen, 2022). These methods can effectively detect sudden changes. However, the data obtained from the liver experiment not only has a sudden change in variance, but also has constant changes in the mean function. This constantly changing mean will affect the detection effect of the variance change point. Directly applying these methods to the liver data will lead to an ironic detection result. For the smooth mean estimation in the liver data, the smoothing spline is one of the common method (Shang & Cheng, 2013; 2017; Xu & Wang, 2018). There are also some methods combining mean function estimation and change detection together (Loader, 1996; Liao & Meyer, 2017; Grégoire & Hamrouni, 2002). However, all these methods are about to estimate the sudden changes in the mean curve. They are not suitable for our scenario, where the mean function changes constantly in our scenario.

Gao et al. (2019; 2020) proposed algorithms that can do the smooth mean function estimation and the variance change point detection simultaneously. However, limited by the practical meaning of the application scenario, it is very common that the location of the change point is normally in a certain range. In the liver experiment, we found that the variance change point of the temperature profile is basically distributed in 10-16 hours. However, the computational logic of their methods needs to scan the entire data profile, in which most parts of the data series have no change point. Therefore, scanning the whole time series to detect variance change points is unnecessary. There are more than 36,000 temperature profiles need to be analyzed in the scene of time competition. Clearly their methods are highly lack of efficiency.

In this paper, we propose a new variance change point detection approach with dual control windows to improve the computational efficiency of the method proposed by Gao et al. (2019) to simultaneously improve the accuracy and speed of the change detection. In our proposed method one control window $CW_\alpha$ is designed for the mean estimation and the other control window $CW_\beta$ is designed for the variance change point detection. In the liver procurement experiment, the variance change point of the liver surface temperature generally conforms to a normal distribution. We obtain the preliminary variance change points information by randomly scanning a group of temperature profiles in the early stage, and construct a model for the mean estimation based on the distribution information of the variance change point. Our proposed method reduces the redundant data information sufficiently and allows to address an accurate estimate of the change point fleetly, which significantly improves the computational efficiency of the algorithm, and saves plenty of time in detection for urgent life saving circumstances.

## 2 METHOD

### 2.1 MODEL AND NOTATION

Suppose that $y_t$ are the independent observations generated from the following model:

$$y_t = f_0(t/n) + \epsilon_t, \quad t = 1, \ldots, n,$$

where $f_0$ is an unknown smoothing function, $\epsilon_t \sim N(0, \sigma_t^2)$ is the error term. Let $\tau$ be the position of the variance change point. $\sigma_0^2$ and $\delta_0^2$ are variances. For $t \leq \tau$, $\sigma_t^2 = \sigma_0^2$. For $t > \tau$, $\sigma_t^2 = \delta_0^2$.

Algorithm 1 is the computational framework of our proposed method. This algorithm begins with an initialization of the dual control windows. To obtain a possible range of the potential variance change points, we firstly perform a preliminary variance change point detection procedure and construct an initialized control window $CW_\alpha^0$ for the mean estimation and a control window $CW_\beta^0$ for the variance change point detection based on the randomly selected $K$ data profiles. The variance change point detection procedures of these $K$ randomly selected data profiles are the method by Gao et al. (2019). Since the sample space of determining the control window $CW_\alpha^0$ will be updated iteratively in Algorithm 1, the choice of $K$ is arbitrary as long as it satisfies the minimum statistical requirement. Then we estimate the mean function and detect the variance change point within the control windows. The dual control windows $CW_\alpha$ and $CW_\beta$ are updated iteratively and simultaneously based on the information of the newly detected variance change points. The details of the dual control windows are shown in Section 2.2.

All parameters' estimates are obtained by minimizing the following objective function,

$$\frac{1}{n}(\mathbf{y} - \mathbf{f})^\top \Sigma_{n,\tau,\sigma,\delta}^{-1}(\mathbf{y} - \mathbf{f}) + \lambda J(f), \tag{1}$$

where $\mathbf{y} = (y_1, y_2, \ldots, y_n)^\top$, $\mathbf{f} = (f(1/n), f(2/n), \ldots, f(1)^\top)$, $\Sigma_{n,\tau,\sigma,\delta}$ is a diagonal matrix, where the first $\tau$ diagonals are $\sigma_0^2$, and the rest are $\delta_0^2$. $J(f)$ is the roughness penalty with the smoothing parameter $\lambda > 0$ balancing the tradeoff between the smoothness of the estimated mean function and the goodness of fit represented by the weighted sum of squared errors. Since the objective function Equation 1 tends to zero as $\sigma^2$ goes to infinity, the global minimizer of Equation 1 does not exist. Therefore an iterative parameter estimation procedure is designed to obtain a local optimal solution. The parameter estimation starts with the initialization of the mean function $\hat{f}^{(0)}$ assuming constant variance. Therefore, the covariance matrix in Equation 1 reduces to $\sigma^2 I$, and $\sigma^2$ can be absorbed into the smoothing parameter $\lambda$. Given the mean function estimate, the variance change point $\hat{\tau}$, and the variances $\hat{\sigma}^2$ and $\hat{\delta}^2$ are obtained in the change detection procedures. The mean function estimates and the detected variance change points are updated iteratively. The convergence criterion of our proposed algorithm is the maximum absolute difference between the residuals of the current iteration and the previous iteration. In the numerical experiments conducted in the later sections, our algorithm can converge in a few iterations. The consistency of parameter estimates has been proved by Gao et al. (2019).

### 2.2 THE DUAL CONTROL WINDOWS

Suppose we have detected $K$ data profiles, and $\{\iota_1, \ldots, \iota_K\}$ is the set of detected variance change points. To build a proper control window $CW_\alpha$ for the mean estimation, we have verified that the asymptotic distribution of the variance change point is normal. According to the empirical rule, about 95% of the data falls into the range of the mean plus or minus two standard deviations. More general, we use the upper quantiles of the standard normal distribution as an approximate substitute to get $CW_\alpha'$.

$$CW_\alpha' = [CLL_\alpha', CLU_\alpha'], \quad CLL_\alpha' = \mu - W, \quad CLU_\alpha' = \mu + W,$$

---

**Algorithm 1:** Variance Change Point Detection with Dual Control Windows

---

**Input:** Data set $\mathscr{D} \in \mathbb{R}^{M \times n}$, where $M$ is the number of data profiles, $n$ is the length of each data profile

**Output:** $\{\hat{\tau}_1, \ldots, \hat{\tau}_M\}$

**Step1** Initialize the dual control windows. Randomly select K data profiles from $\mathscr{D}$, detect the variance change points $\hat{\iota}_1, \ldots, \hat{\iota}_K$ to initialize the control window $CW_\alpha^{(0)}$ and $CW_\beta^{(0)}$.

**Step2** Variance change detection within dual control windows. For the $m^{th}$ data profile,

    **Step2.1** capture the reduced data profile $\mathbf{y}_m^* = \left\{ y_i | i \in CW_\alpha^{(m-1)} \right\}$.

    **Step2.2** Initialize the mean function estimate $\hat{f}_m^{(0)}$ assuming constant variance. $\hat{f}_m^{(0)}$ minimizes $\frac{1}{n}(\mathbf{y}_m - \mathbf{f}_m)^\top (\mathbf{y}_m - \mathbf{f}_m) + \lambda J(f_m)$.

    **Step2.3** Estimate the variance change point iteratively. Each iteration consists of two steps. At the $j^{th}$ iteration,

        **Step-A** Given the mean estimate $\hat{f}^{(j-1)}$, the data profile $\mathbf{y}_m^{**} = \left\{ y_i | i \in CW_\beta^{(m-1)} \right\}$ is validated to yield estimates of $\hat{\tau}^{(j)}$, $[\hat{\sigma}^2]^{(j)}$ and $[\hat{\delta}^2]^{(j)}$.

        **Step-B** Substitute the current parameters estimates from Step-A into Equation 1 to update the mean function estimate $\hat{f}^{(j)}$.

    **Step2.4** Iterate until the algorithm converges, and obtain the estimates $\hat{\tau}_m, \hat{\sigma}_m^2, \hat{\delta}_m^2$, and $\hat{f}_m$.

    **Step2.5** Update $CW_\alpha^{(m)}$ and $CW_\beta^{(m)}$ by $\{\iota_1, \ldots, \iota_K, \hat{\tau}_1, \ldots, \hat{\tau}_m\}$.

---

where $\mu = \frac{1}{K} \sum_{i=1}^{K} \iota_i$, $W = z_q \sqrt{\frac{1}{K-1} \sum_{i=1}^{K} (\iota_i - \mu)^2}$, $K$ is the number of detected change points, $\tau$ is the known change point data, and $z_q$ is the upper quantile of the standard normal distribution corresponding to the desired accuracy.

In our method, we use the cubic splines method to estimate the mean functions. The computational efficiency of the smoothing spline method is highly affected by the sample size of the data. The minimizer $f_\lambda$ of Equation 1 resides in the $n$-dimensional space, and the computation in multivariate settings is generally of the order $O(n^3)$ (Kim & Gu, 2004). Therefore, large sample size will extremely slow down the computational speed. However, the accuracy of the mean function estimation using the nonparametric method highly relies on the sufficiency of the quantity of the data. In order to improve the speed of the algorithm without reducing the accuracy of parameter estimates, we need to choose an appropriate radius of the control window $CW_\alpha$. We define a hyper parameter $W_\alpha$, which is the minimum size of the control window $CW_\alpha$. The determination of $W_\alpha$ refers to the rule of elbow in the K-means clustering. The minimum window size can be estimated by balancing the tradeoff between the accuracy of the estimation and computational speed. For example, in the liver experiment, if we want the accuracy of the change detection to be above 0.95, we should ensure that the control window contains at least 100 data points. In that case, to make the calculations as fast as possible, we can choose $W_\alpha = 50$. Considering the above two factors, the definition for $CW_\alpha$ is

$$CW_\alpha = [CLL_\alpha, CLU_\alpha], \ CLL_\alpha = \mu - \max\{W_\alpha, W\}, \ CLU_\alpha = \mu + \max\{W_\alpha, W\},$$

where $\mu = \frac{1}{n} \sum_{i=1}^{n} \tau_i$, and $W = z_q \sqrt{\frac{1}{n-1} \sum_{i=1}^{n} (\tau_i - \mu)^2}$.

$CW_\beta$ is designed for variance change point detection. Generally, the mean function estimation requires more data information than the variance change points detection. Therefore, a portion of the data contained in $CW_\alpha$ is redundant for variance change detection. Generally speaking, the change point location presents a high degree of concentration. This is a common situation in many change point detection problems, such as the change of the daily traffic flow data set of the same road section, the change of the stock market, and the change of the sales of an industry company.

148 Therefore, the variance change point detection interval $CW_\beta$ should be further controlled to make
149 the probability of variance change points in the interval reaches to a certain value or more, and
150 those points that are not likely to become variance change points are eliminated as much as possible.
151 According to the Glivenko-Cantelli Theorem, we have

$$P\left\{\lim_{m\to\infty}\sup_{-\infty<\tau<\infty}|F_m(\tau)-F(\tau)|=0\right\}=1,$$

152 where $F_m(\tau)$ is the empirical distribution function of the dataset $\{\tau_1,\ldots,\tau_m\}$ and $F(\tau)$ is the real
153 distribution function of the variance change points. When the sample size is large enough, we adopt
154 the concepts of the probability interval to control the likelihood of the change point positions ac-
155 cording to the empirical distribution function. We get $CW_\beta=[CLL_\beta,CLU_\beta]$, where the potential
156 variance change points fall into this interval with a probability of at least $q$. The upper and lower
157 bounds of $CW_\beta$ satisfy $F_m(CLU_\beta)-F_m(CLL_\beta)\geq q$. Since the asymptotic distribution of the
158 variance change point is normal, we can do it in terms of the quantile of the sample

$$CW_\beta=[CLL_\beta,CLU_\beta],\quad CLL_\beta=\tau_{\frac{1-q}{2}},\quad CLU_\beta=\tau_{\frac{1+q}{2}},$$

159 where $\tau_{\frac{1\pm q}{2}}$ is the $\frac{1\pm q}{2}$ percentile of the detected variance change points. By using the control
160 window $CW_\beta$, we further reduce the amount of data used for variance change point detection.

### 2.3 SMOOTHING SPLINES ESTIMATION FOR MEAN FUNCTION

162 The mean estimation in this article uses the cubic smoothing splines. Before introducing the es-
163 timation method for the mean function, it is necessary to first introduce the polynomial smooth-
164 ing spline with period 1 on the interval $[0,1]$. The mean function $f_0$ is estimated by minimiz-
165 ing the objective function $\frac{1}{n}\sum_{i=1}^{n}\left(y_i-f_0(t_i)\right)^2+\lambda\int_0^1\left(f_0^{(m)}(t)\right)^2dt$ in the space $\mathcal{C}^{(m)}[0,1]=$
166 $\{f:f^{(m)}\in\mathcal{L}_2[0,1]\}$, where $\mathcal{C}^{(m)}[0,1]$ is a reproducing kernel Hilbert space (RKHS). The true
167 mean function $f_0$ is an unknown smoothing function and is a function in the reproducing kernel
168 Hilbert space $\mathcal{C}=\{f\mid f:[0,1]\to R,J(f)<\infty\}$. When $\sigma^2,\delta^2$ and $\tau$ are given, the estimate of $f_0$
169 is the minimizer of the penalized weighted least squares Equation 1, where the smoothness param-
170 eter $\lambda>0$ is chosen by the generalized cross-validation. Note that since $\mathcal{C}$ is of infinite dimension,
171 it is not possible to optimize Equation 1 on $\mathcal{C}$ directly. However, since the weighted least squares
172 part in Equation 1 depends on $f_0$ only at the observation $y_i,i\in[CLL_\alpha,CLU_\alpha]$, the representation
173 theorem guarantees that the exact minimizer exists in a finite dimensional subspace of $\mathcal{C}$. Therefore,
174 the minimizer of the objective function Equation 1 can be analytically obtained.

### 2.4 VARIANCE CHANGE POINT DETECTION

176 When the mean function estimate $\hat{f}$ is given, the variance change point $\hat{\tau}$ is detected within the
177 control window $CW_\beta$ through a well designed hypothesis test procedure. The estimates of the
178 variances are $\hat{\sigma}^2=\hat{\tau}^{-1}\sum_{t=1}^{\hat{\tau}}\left\{y_t-\hat{f}(t/n)\right\}^2$ and $\hat{\delta}^2=(n-\hat{\tau})^{-1}\sum_{t=\hat{\tau}+1}^{n}\left\{y_t-\hat{f}(t/n)\right\}^2$. For
179 variance change point detection, establish the null and alternative hypotheses.

$$H_0:\sigma_1^2=\cdots=\sigma_n^2\quad vs\quad H_1:\sigma_1^2=\cdots=\sigma_\tau^2\neq\sigma_{\tau+1}^2=\cdots=\sigma_n^2$$

180 For a potential variance change point $\hat{\tau}=k,k\in\{1,\ldots,n\}$, the likelihood function is

$$L(k)=k\left[\log\frac{1}{k}\sum_{t=1}^{k}\left\{y_t-\hat{f}(t/n)\right\}^2\right]+(n-k)\left[\log\frac{1}{n-k}\sum_{t=k+1}^{n}\left\{y_t-\hat{f}(t/n)\right\}^2\right],$$

181 then we have $L(n)=-2L_0(\hat{\sigma}^2)-n-n\log 2\pi$, and $L(\tau)=-2L_1(\hat{\sigma}^2,\hat{\delta}^2)-n-n\log 2\pi$. Where
182 $L_0$ and $L_1$ are the log-likelihood functions under the hypotheses $H_0$ and $H_1$. Then we have the
183 definition of the test statistic: $\Delta_n^2=\max_{1<k<n}\{L(n)-L(k)\}$. By the principle of the minimum

information criterion, there is no evidence of the existence of the variance change point within the control window $CW_\beta$ if $L(n) \leq \min_k L(k)$, $k \in [CLL_\beta, CLU_\beta]$. The null hypothesis is failed to reject if $\exists k \in [CLL_\beta, CLU_\beta]$, we have $L(n) > L(k)$, then the null hypothesis is rejected, and there exists a variance change point. Therefore the position of the estimated variance change point is

$$\hat{\tau} = \underset{CLL_\beta \leq k \leq CLU_\beta}{\arg\min} L(k).$$

# 3 SIMULATION

In this section, we compare the performance of our proposed change detection method, which is denoted by the New method hereafter, with one of the most sufficient method in Gao et al. (2019) hereafter denoted by the Gao19 method. We conduct the comprehensive simulation studies using the same data generating schemes in Gao et al. (2019). Two mean functions $f_{01}(t) = 20 + 12t(1-t)$ and $f_{02}(t) = sin(t) + t^5 - 8t^3 + 10t + 6$ are considered. The first mean function $f_{01}$ mimics the trend of the temperature profile obtained in the porcine liver procurement experiment, while the second function $f_{02}$ represents a more complex smooth mean trend. We use two sample sizes $n = 130$ and $n = 500$. When $n = 130$, the true variance change point is $\tau_0 = 65$ and the parameter of the control window $W_\alpha = 50$. When $n = 500$, we have $\tau_0 = 250$ and $W_\alpha = 190$. The true variances are $\sigma_0^2 = 0.24$ and $\delta_0^2 = 0.06$ when $f_{01}$ is the mean function, and $\sigma_0^2 = 8$ and $\delta_0^2 = 2$ when $f_{02}$ is the mean function. We simulated 1000 data replicates for each combination of simulation settings. We also conduct the sensitivity and power analysis of the proposed method. However, due to the limitation of the space, we include these studies in Appendix.

Figure 2 is the boxplots of the change point estimates of the proposed algorithm and the method of Gao19. New is our proposed algorithm. Gao19 and New are tested with sample size $n = 130$. New500 represents the change point estimates of the proposed algorithm with sample size $n = 500$. We can see that the accuracy of the change estimates is the same for the proposed method and Gao19. However, the number of the extreme change estimates is greatly reduced by the proposed method. This is due to the fact that Gao19 identifies pseudo change points far from the actual position of the change point if there are strong abnormal data fluctuation behaviors near the boundaries of the data profiles. The new algorithm, however, restricts the data to the interval where the change points have a high probability of existence, which reduces the influence of the extreme pseudo change points and thus reduces the extreme change estimates. When the sample size increases from 130 to 500, the estimation accuracy is significantly improved and the extreme change estimates are further reduced, which indicates that the proposed method has a better effect on improving the accuracy of the change estimates.

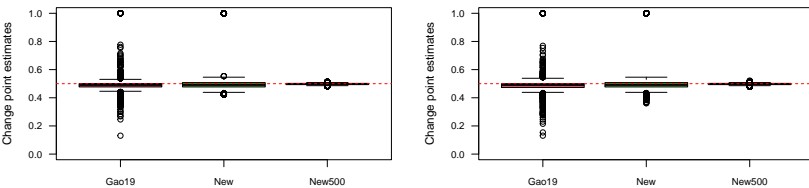

Figure 2: Boxplots of the change point estimates. The left panel is the simulation result with the true mean function $f_{01}(t)$; the right side panel is the result with $f_{02}$. Both methods are tested with sample size $n = 130$. New500 represents the change point estimates of the proposed algorithm with sample size $n = 500$. The red dashed line is the true change point $\tau_0/n = 0.5$.

To compare the computational efficiency of our proposed algorithm with Gao19, we conduct simulations with the mean function $f_{01}$ and the sample size $n = 130$. We generate $m = 5,000$ and $m = 50,000$ replicates. The experiments are repeated for five times, and the average results are shown in the Table 1. We can see that the average time consumed by the method of Gao19 is 280.92s for $m = 5,000$ data profiles, and is only 156.96s for our proposed algorithm. This is a

220 significant improvement in computational speed. The difference of the time consumption for these
221 two methods are linearly increased as the amount of data increases. Combining the results of Fig-
222 ure 2 and Table 1, we can draw a conclusion that our proposed algorithm achieves a significant
223 improvement in computational efficiency on the basis of guaranteed estimation accuracy.

Table 1: Computational efficiency Comparison.

| Method | Gao19 | New |
|---|---|---|
| m=5,000 | 280.92s | 156.96s |
| m=50,000 | 2,634.93s | 1,601.53s |

224 Figure 3 evaluates the performance of the mean estimates of the proposed method. We compute
225 the mean squared errors and plot the boxplots. The left side panels show the 25th, 50th, and 75th
226 percentiles of the MSEs for sample sizes $n = 130$ and $500$. We can see that within the control
227 window $CW_\alpha$, the mean function estimates match the true mean function well. In addition, the
228 MSE of the mean estimation improves as the sample size increases.

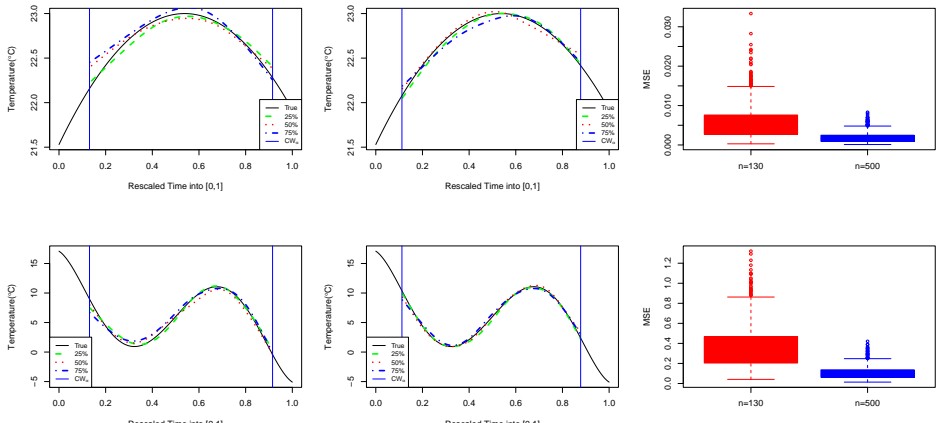

Figure 3: Performance of the mean estimation. The top panels show the simulation results when the
mean function is $f_{01}$. The bottom panels show the simulation results when the mean function is $f_{02}$.
The sample size of the left side panels is $n = 130$ whose MSE are the 25th (dashed green), 50th
(dotted red), and 75th (dot-dashed blue) percentiles of the 1,000 MSEs obtained in each setting. The
middle panels are the same as the left side panels but with $n = 500$. The vertical blue lines are the
upper and lower boundaries of the control window $CW_\alpha$. The right side panels are the boxplots of
the 1,000 MSEs in each setting.

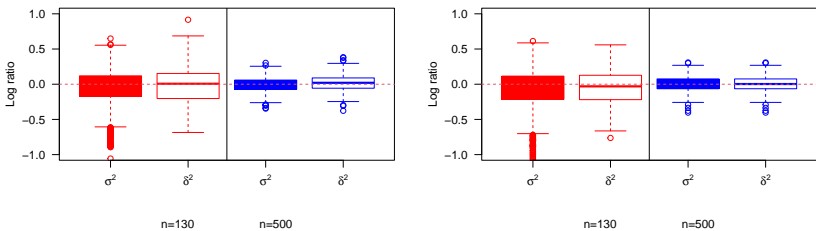

Figure 4: Boxplots of the log ratios of the variance estimates versus the true variances. The left panel
shows the simulation results when the mean function is $f_{01}$. The right panel shows the simulation
results when the mean function is $f_{02}$. Red: $n = 130$; Blue: $n = 500$.

Figure 4 uses the log ratio of the variance estimates versus the true variances to evaluate the estimation performance for the two variances. We can see that both variances are estimated accurately. The estimation performances are improved as the sample size increases from 130 to 500.

## 4 APPLICATION

The data were collected through a well designed noninvasive bimedical experiment conducted by a research team from Virginia Tech. In the mechanical perfusion process, they measured the surface temperature of each profile over 24 hours using a noninvasive, high precision thermal imaging system. They divided a porcine liver's surface into a dense grid composed by 36,795 spots. Temperature measurements were collected every 10 minutes producing a 24 hours surface temperature profile with 145 points in each profile. We discarded the data in the first 2.5 hours, since the perfusion fluid needs one to two hours to completely infuse and stabilize the liver. Finally, there were 130 points left in each profile. Previously, (Gao et al., 2019) conducted a spot-wise analysis method on each of these 36,795 temperature profiles and obtained a heat map of the estimates of the variance change points on the liver surface. We repeat their experiments, and compare their results (Gao19) with our proposed method (New).

Figure 5 is the heat maps of the variance change point detection results of Gao19 and our proposed method. Different color indicates that different areas of the organ have different viability deterioration time. An earlier change in variance means an earlier drop in the viability of the cells around the spot. Two maps share the similar color patterns of regional hierarchical structure, which is the viability of the top half and the middle bottom parts of the liver deteriorated around 12 hours while the left and right bottom corners of the liver last beyond 14 hours. There are also several clearly visible straight green line type boundaries between the early and late failure areas. These may be parts where the porcine liver lobe was deformed during dissection and perfusion. The detection results of Gao19 have been validated by the biomedical scientists. Similar regional color patterns of these two heat maps suggest that the results of our proposed algorithm are accurate in a practical sense. However, there are still macroscopic differences. In the bottom left and right areas of the heat map, there are less green dots of our New method. That means our proposed algorithm discovers more earlier deteriorated change points distributed around 12 hours than Gao19. Our results are more consistent with the biological and biomedical conclusions. We also compare the computational efficiency of these two methods. The computational time of Gao19 is 3,589.66s, while the new algorithm only takes 2,099.45s. The proposed method can greatly reduce the detection time. This makes the online organ viability assessment in realtime possible.

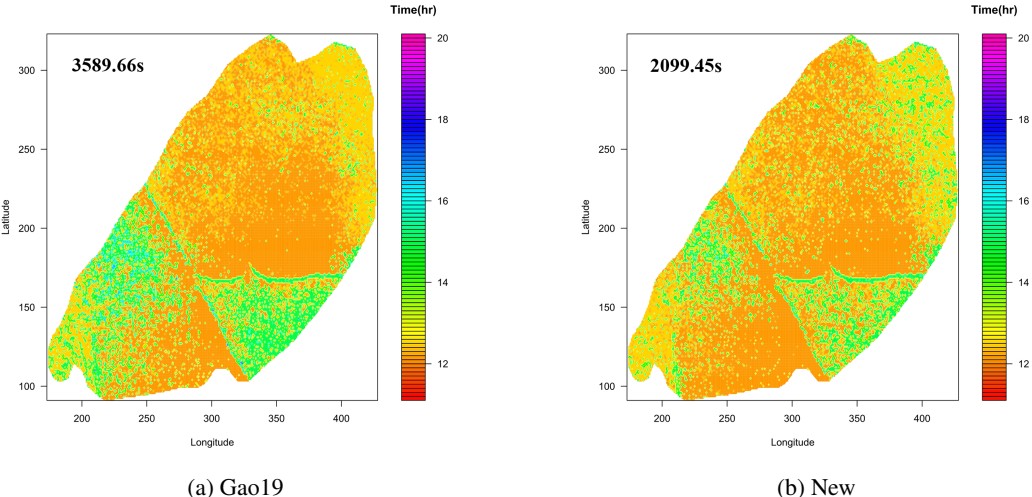

(a) Gao19

(b) New

Figure 5: Heat map of the estimated variance change points on the surface of porcine liver.

Figure 6 is the mean function and variance change point estimates of our proposed method applied respectively on the raw and de-trended temperature profiles at three randomly selected spots on the

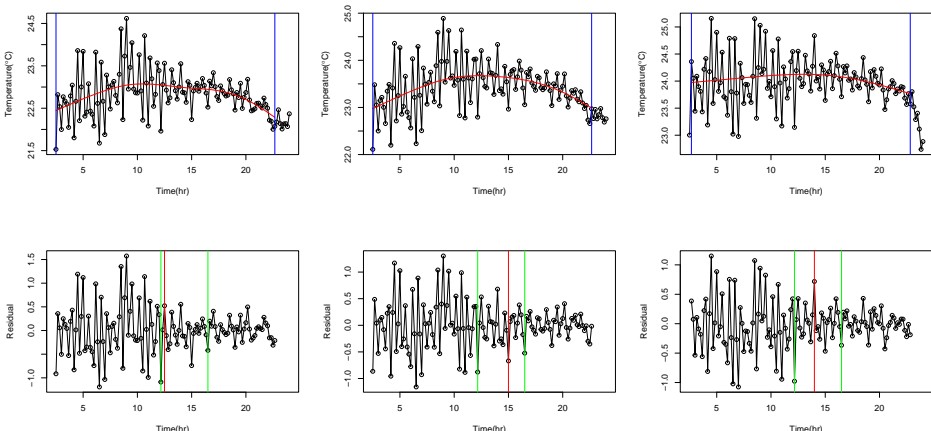

Figure 6: Mean and variance change point estimates imposed respectively on the raw temperature profiles (top panels) and de-trended temperature profiles (bottom panels) at three randomly selected spots on the liver surface. The smooth red curves are the estimated mean function. The blue vertical lines are the upper and lower bounds of the control window $CW_\alpha$. The green vertical lines are the upper and lower bounds of the control window $CW_\beta$. The red vertical lines are the estimated variance change points.

liver surface. All mean estimates fit well with the potential trends hidden in the data. All detected variance change points reasonably locate at the proper locations in the data profiles. In the first 12 hours or so, the average temperature at these three spots rises at different speeds and has a faster downward trend after 12 hours. The variance change points of these three spots are all around 12 hours, which is consistent with the biomedical conclusion.

## 5 CONCLUSION

The viability detection of the transplanted organs is an important biomedical issue. In organ transplantation, timelines are the most important thing. The noninvasive viability change detection algorithm proposed by Gao et al. (2019) can well solve the existing problems and provide a reasonable viability assessment of the organ. However, their method spends a lot of time to explore the areas where change points unlikely exist, thus losses the computational efficiency. It is necessary to improve the computational efficiency of the algorithm. Motivated by this, we propose an evolutionary algorithm with a well designed double-layer control windows to filter out these noninformative data. The simulation shows that the proposed algorithm reduces the computational time significantly without losing detection accuracy. In the application, the heat map of the detected viability change points on the organ surface obtained by the proposed algorithm discovers earlier deteriorated areas on the liver surface, which is more consistent with the biomedical conclusions. The proposed change detection method has a very high application value in the field of online change detection.

## AUTHOR CONTRIBUTIONS

## ACKNOWLEDGMENTS

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

APPENDIX

## A    SENSITIVITY ANALYSIS OF THE METHOD WITH RESPECT TO THE SAMPLE SIZE

The performance of our proposed change detection method is affect by the sample size of the data profile. Here we perform the test on the data with mean function $f_{01}(t)$. The sample size $n$ is set to be $\{20, 25, 30, 35, \ldots, 130\}$. The true variance change point $\tau = \frac{n}{2}$. The variances before and after the change point are $\sigma_0^2 = 0.24$ and $\delta_0^2 = 0.06$. We simulate 10,000 data profiles for each sample size. The change point estimation accuracy and the computational efficiency of the proposed algorithm are shown in Figure 7. We can see that the accuracy of the proposed algorithm increases rapidly as the sample size increases. The accuracy has a steeper curve before $n = 100$ and slows down afterwards. The computational cost is linearly related to the sample size. According to the rule of elbow, we choose $n = 100$ in the mean function estimation stage, which makes the minimum window size $W_\alpha = 50$. By choosing $n = 100$ the accuracy of the proposed algorithm exceeds 90%, and the time consuming is about 400 seconds per 10,000 data profiles.

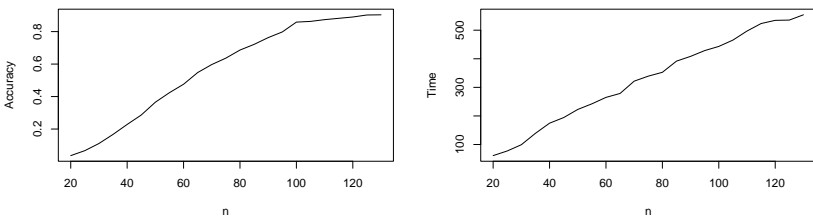

Figure 7: The accuracy and computational speed of the proposed method against the sample size.

## B    POWER ANALYSIS

We also present a power analysis study on the change point testing procedure. We considered the data simulated from the mean function $f_{01}(t)$. The variance $\delta_0^2 = 0.06$ and $\sigma_0^2 = \theta \delta_0^2$, where $\theta \geq 1$ was the ratio of $\sigma_0^2$ over $\delta_0^2$.

When $\theta = 1$, $\delta_0^2 = \delta_0^2$ indicates there is no variance change point in the data profile. Therefore we can investigate the size of the test under this setting. Two levels $\alpha = \{0.05, 0.1\}$ and four sample sizes $n = \{130, 500, 2000, 10000\}$ are considered. For each combination of $\alpha$ and $n$, we simulated 10,000 replicated data profiles. Table 2 summarizes the results about the size of the test. We see that the empirical sizes of the test are smaller than the levels of the test indicating that the test is a bit more conservative than expected in claiming a change point when there is no change point. The reason is that the asymptotic null distribution is a heavy-tailed extreme value distribution and may require a larger sample size to achieve the desired size of the hypothesis test.

Table 2: Proportions of rejections when $H_0$ is true under different levels of $\alpha$. The true mean function is $f_{01}(t)$.

| Test Level | Sample Size $n$ | | | |
|---|---|---|---|---|
| $\alpha$ | 130 | 500 | 2000 | 10000 |
| 0.05 | 0.0000 | 0.0002 | 0.0010 | 0.0107 |
| 0.1 | 0.0325 | 0.0336 | 0.0356 | 0.0401 |

When $\theta > 1$, we investigate the power of the proposed change detection method in the simulation. The power plots are shown in Figure 8. Consider the data generated with the mean function $f_{01}(t)$. The variances $\delta^2 = 0.06$ and $\sigma^2 = \theta \delta^2$, where $\theta \geq 1$ is the ratio of $\sigma^2$ over $\delta^2$. Two sample sizes $n = 130$ and 500 are considered. When $n = 130$, $\theta$ takes the value of $\{1, 1.25, 1.5, \ldots, 5\}$. When $n = 500$, the grid points of $\theta$ are $\{1, 1.2, 1.4, \ldots, 5\}$. For each combination of $\theta$ and $n$, we simulate 10,000 data replicates. For $n = 130$, the power is greater than 0.8 when the variance ratio is 3. For $n = 500$, the power is higher than 0.9 before the variance ratio reaches 2.

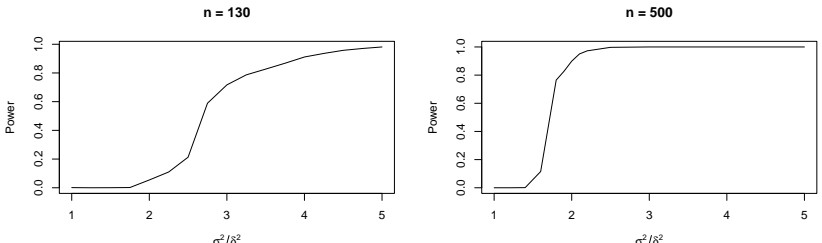

Figure 8: Plots of detection power against the ratio of the variances.

