# OpenReview forum: "A Effective Variance Change Detection Method under constantly Changing Mean"
_ICLR.cc/2024/Conference — ICLR 2024 Conference Withdrawn Submission_

### Official Review · Reviewer_dHKn · 2023-10-31

**Soundness:** 2 fair
**Presentation:** 2 fair
**Contribution:** 3 good
**Rating:** 5
**Confidence:** 4

**Summary:**

The viability detection of the transplanted organs is an important biomedical issue. In organ transplantation, timelines are the most important thing. Gao19 proposed a well-designed noninvasive method evaluated the viability status of organs by detecting the variance change point of their surface temperature through exploring the entire data profile. However, most of the data in a temperature profile are redundant, which yields a waste of computational resources of their method. This paper proposes an accelerating algorithm with a well-designed dual control windows scheme that can be extended to online change detection. The proposed method significantly improves the computational speed and retains the same change detection power as the method Gao19. Simulation and application results demonstrate the robust performance of the proposed method.

**Strengths:**

1. In terms of originality, this paper proposes a new accelerating algorithm and designs a dual control window scheme, which eliminates redundant data for mean estimation and variance change point detection, preserves information data, and improves computational efficiency.
2. In terms of quality, this paper is technically correct, experimentally rigorous and reproducible.
3. In terms of clarity, this paper clearly describes the motivation, notation, details of the dual control window scheme, model, and algorithm.
4. In terms of significance, this paper focuses on viability detection of the transplanted organs and provides an online version based on Gao19 method, which reduces detection time. The proposed change detection method has a very high application value in the field of clinic.

**Weaknesses:**

1.The core of this paper is that it improves the computational efficiency of the algorithm, so it is necessary to supplement the theoretical proof of convergence speed and compare it with the Gao19 method to make the new method more convincing.
2. This article lacks novelty and is an improvement on the Gao19 method. The mean estimation method and variance change point detection method, as well as the experimental design, are the same as the Gao19 method.

**Questions:**

1.There are some errors in the details, such as the mixing of upper and lower case N. It is recommended to modify t in the test statistic to k. Reversed the numerator and denominator of the ratio defining the parameter theta. There are also some professional terms such as ‘hypothesis’ instead of ‘assumption’ on page 5. It is necessary to explain the use of "eta" in the objective function.
2.The paper mentions algorithm convergence and needs to introduce which convergence criterion to use.
3. The representation of 'y_i' in the algorithm is incorrect and needs to be modified.
4. This paper pre executed a preliminary variance change point detection program to obtain a set of change points. Please provide a detailed explanation of the method used to obtain this set.

---

> ### Author Response · Authors · 2023-11-15
>
> Thank you very much for all the valuable questions.
>
> To weaknesses:
>
> 1&2. The method presented in this paper focuses more on methods rather than theoretical. Besides, we integrated the dual control window structures into the method of Gao et al. (2019) to sufficiently narrow down the detection range of the variance change points. Therefore, all the theoretical properties of Gao et al. (2019) still apply to our method.
> 2. To conduct a fair comparison, we reproduced the data analysis experiments in Gao19 to illustrate that our proposed method has obvious improvement in both simulation and application.
>
> To questions:
> 1. revised;
> 2. revised and marked in blue, page3;
> 3. revised and marked in blue, page4;
> 4. revised and marked in blue, page3;

---

### Official Review · Reviewer_Cj8H · 2023-11-01

**Soundness:** 2 fair
**Presentation:** 2 fair
**Contribution:** 1 poor
**Rating:** 5
**Confidence:** 4

**Summary:**

A previous JASA paper (Gao et al., 2019) has proposed a method for variance change point detection, with application to studying the surface temperature of transplanted organs and their viability. Motivated by the high computational time and complexity of this method, this paper proposes an approach to speed up the computation through uses of dual control windows. Numerical experiments are presented parallel to the original paper (Gao et al., 2019).

**Strengths:**

1. The paper has a good motivation
2. Description of the application is clear

**Weaknesses:**

1. The contribution is very limited. Except from the proposed dual control window in section 2.2, all of the rest contents are from the paper (Gao et al., 2019).
2. Using the dual control window only speeds up the computation time by 40%, but incurs quite significant estimation bias compared to the result in Gao et al. (2019), as show in the application section.
3. The algorithm for dual control windows is not well described and the details are a bit confusing, e.g. regarding the iterations over data profile index m and the iteration j in Algorithm 1.
4. The test statistic \Delta_n is shown in Gao et al. (2019) to asymptotically follow an extreme value distribution, which is how a rejection criterion can be chosen with control of the type I error. This is not included in this paper, and the proposed rejection criterion in section 2.4 could have a really high type I error.

**Questions:**

See weaknesses.

As discussed in section 2.1, the objective (1) doesn't have a global minimizer since \sigma and \delta can go all the way to infinity, so only a local minimizer is searched for. While I am aware that the same objective is used in Gao et al. (2019), I wonder why not to use the objective taking the form of penalized log-likelihood of a Gaussian model, such that \log\det(\Sigma) is added to the objective and global minimizers now exist?

---

> ### Author Response · Authors · 2023-11-15
>
> Thank you very much for all the valuable questions.
>
> To weaknesses:
> 1. The method presented in this paper integrated the dual control window structures into the method of Gao et al. (2019) to sufficiently narrow down the detection range of the variance change points, which significantly improved the change detection efficiency and accuracy. Although this article follows the methodological framework of Gao19, we should also emphasize that the contribution of the proposed method is valuable and sufficient.
> 2. The application is an unsupervised learning environment; we cannot tell which method provided more accurate results. However, as we have motioned in this paper, our proposed algorithm discovers more earlier deteriorated change points distributed around 12 hours than Gao19. Therefore, our detection results are more consistent with the biological and biomedical conclusions. page8.
> 3. I have rearranged the expression of Algorithm 1. page4;
> 4. This is a very good question. We conducted additional simulation studies for type-I error, the results is updated in the Appendix-B. From Table-2, we see that the empirical sizes of the test are smaller than the levels of the test indicating that the test is a bit more conservative than expected in claiming a change point when there is no change point. The reason is that the asymptotic null distribution is a heavy-tailed extreme value distribution and may require a larger sample size to achieve the desired size of the hypothesis test.
>
> To questions:
> This is a very good question. My team is actually working on this direction. But it has not been done because the algorithm is more complex, and hopefully we can report this in the near future. On the other hand, we should also emphasize that the current results are already very good.

---

> > ### Comment · Reviewer_Cj8H · 2023-11-21
> >
> > Thank you to the authors for the reply. I am not very convinced on the contribution of this paper, but several of my questions and concerns have been addressed. I have increased my rating accordingly.

---

> > > ### Author Response · Authors · 2023-11-21
> > >
> > > I am glad that I have addressed some of your valuable questions. Thank you very much for increasing your rating accordingly. The following statement may further illustrates the valuable contribution of our work.
> > >
> > > The major contribution of this paper is significant to the biomedical meanings. Whether or not we can accurately evaluate the viability of the procured organs is extremely important. It is a life saving work. Organ transplantation is an effective method to treat end-stage diseases of various organs. According to the China Organ transplantation Development Report, more than two millions people around the world need to receive organ transplants every year, with an average organ supply/demand ratio around 1/30 to 1/20. In 2020, China completed 17,897 organ transplants, which has increased compared with previous years, but there is still a huge gap for demand. Each procured organ is extremely valuable. In organ transplantation, the life-support organs, such as the heart and liver, are basically obtained from brain-dead organ donors who meet the criteria of neurological death, and there are great uncertainties in time and space. From obtaining an organ to match it to a suitable recipient, it often takes hours and even days. Due to the limitation of storage and transportation techniques, the viability of organs cannot be fully guaranteed. The quality of procured organs must be evaluated before transplantation.
> > >
> > > As we have shown in the application section, our proposed method has discovered more earlier deteriorated change points distributed around 12 hours than Gao19. Therefore, our detection results are more consistent with the biological and biomedical conclusions. This is very significant and valuable improvement to the organ procurement and transplantation.

---

### Official Review · Reviewer_daaU · 2023-11-03

**Soundness:** 3 good
**Presentation:** 2 fair
**Contribution:** 2 fair
**Rating:** 5
**Confidence:** 3

**Summary:**

This article describes a procedure to detect variance changes in surface temperature signals of organs. The authors assume that the signal mean is a smooth function, which makes the task more difficult. The proposed approach is a fast extension of [1] and [2]. Roughly, the authors restrict the number of candidate change-point indexes of the original method.

The original method simultaneously estimates the smooth mean function and the potential variance change-point index. The proposed algorithm splits the two operations. In addition, the authors introduce two "control windows": one limits the number of indexes on which change detection is performed, and the other limits the number of indexes on which the mean estimation is performed.

[1] Zhenguo Gao, Zuofeng Shang, Pang Du & John L. Robertson (2019) Variance Change Point Detection Under a Smoothly-Changing Mean Trend with Application to Liver Procurement, Journal of the American Statistical Association,114:526,773-781, DOI: 10.1080/01621459.2018.1442341

[2] Zhenguo Gao, Pang Du, Ran Jin, John L. Robertson "Surface temperature monitoring in liver procurement via functional variance change-point analysis," The Annals of Applied Statistics, Ann. Appl. Stat. 14(1), 143-159, (March 2020)

**Strengths:**

The algorithm solves a complex task with an interesting application. The fact that is can be applied to real data is also a strength.

**Weaknesses:**

The main objective of the article is to describe a fast alternative to an already existing method [1, 2]. The authors should better explain the complexity gain of their algorithmic contribution. Without this, the processing time improvement could merely result from a better implementation.

If there are other contributions, the authors should better highlight them.

**Questions:**

- Thanks to the proposed approach, the processing time of a thermal image of a porcine liver decreases from 1h to 30 minutes. I cannot tell if this improvement is significant as I am unfamiliar with this application. If 30 minutes is an acceptable execution time, could it be obtained by simply parallelizing the current algorithm? Since each pixel of the thermal video is processed independently, a 3x or 4x speed-up is expected on a personal laptop.

- It is not clear to me if the variance change point detection procedure (Section 2.4) is a contribution of this work. Can you clarify?

- The new algorithm seems more accurate because the candidate change points are restricted to a window. This can prevent false detection on the edges of the signal. How is your method better than a post-processing of the change-points of Gao19 (e.g., by computing a window on the results of Gao19)?

Minor comments:
- In the title: "An Effective Variance Change Detection Method under Constantly Changing Mean".
- L124: "Large sample size will slow down the computational speed." This statement is vague. You could provide the computational complexity, for instance.
- How are the initial change-points estimated at the beginning of the algorithm (Step 1)?
- "Equation 1" instead of "equation 1".
- typo L251
- typo L178
- typo L113
- typo L116-117
- More typos.

---

> ### Author Response · Authors · 2023-11-15
>
> Thank you very much for all the valuable questions.
>
> To questions:
> 1. As mentioned in the first paragraph of this paper, in an organ transplantation operation, both doctors and patients are racing against time. Even 30 seconds is a luxury, not to mention 30 minutes. If we run our proposed method and Gao19 method together on the same computer, ours is constantly faster than Gao 19.
> 2. The variance change point detection procedure is the step-A of our proposed algorithm, page4;
> 3. The method presented in this paper integrated the dual control window structures into the method of Gao et al. (2019). The proposed method sufficiently narrowed down the detection range of the variance change points, which significantly improved the change detection efficiency and accuracy.
>
> To minor comments:
> 1. revised;
> 2. revised and marked in blue, page4;
> 3. revised and marked in blue, second paragraph on page3;
> 4. revised;
> 5-9: revised.

---

> > ### Comment · Reviewer_daaU · 2023-11-21
> >
> > Many thanks for your answers.
> >
> > You have addressed several of my concerns.
> >
> > Nevertheless, can you provide a quick complexity comparison between the new method and Gao19, to better demonstrate that the new algorithm is faster? Since this is the main claim of the paper, it would greatly help if this fact was more rigorously explained.

---

> > > ### Author Response · Authors · 2023-11-22
> > >
> > > Thanks for your question.
> > > 1. The numerical studies in this paper (pape 7, Table-1) provided the evidence of high efficiency of our method. Generally our method reduced 40%~45% of computational time than Gao19.
> > > 2. The computational complexity of the variance change detection procedure (Step-A in Algorithm-1) is O((log n)^4(log log n)^2), and is O(n^3) for the mean estimation procedure (Step-2.2 and Step-A in Algorithm-1) (For more details of the relative theoretical properties please see Kim&Gu 2004 and Gao et al. 2019). Sufficient reduction in sample size will increase the computational efficiency significantly.

---

### Meta-Review · Area_Chair_EeBR · 2023-12-05

**Metareview:**

All reviewers agreed that there are several weaknesses, ranging from unclear novelty to limited contributions to open questions regarding the experimental validation. I fully agree with most of the statements made by the reviewers, and most of these concerns raised could not be addressed in a fully convincing way during the rebuttal and disscussion phase. Therefore I recommend rejection of this paper.

**Justification For Why Not Higher Score:**

Thre are simply too many open questions, even after the discussion phase.

**Justification For Why Not Lower Score:**

N/A

---

### Decision · Program_Chairs · 2024-01-16

Reject